# CD8 T Cell Score as a Prognostic Biomarker for Triple Negative Breast Cancer

**DOI:** 10.3390/ijms21186968

**Published:** 2020-09-22

**Authors:** Masanori Oshi, Mariko Asaoka, Yoshihisa Tokumaru, Li Yan, Ryusei Matsuyama, Takashi Ishikawa, Itaru Endo, Kazuaki Takabe

**Affiliations:** 1Breast Surgery, Department of Surgical Oncology, Roswell Park Comprehensive Cancer Center, Buffalo, NY 14263, USA; masa1101oshi@gmail.com (M.O.); 590mariko@gmail.com (M.A.); Yoshihisa.Tokumaru@roswellpark.org (Y.T.); 2Department of Gastroenterological Surgery, Yokohama City University Graduate School of Medicine, Yokohama 236-0004, Japan; ryusei@yokohama-cu.ac.jp (R.M.); endoit@med.yokohama-cu.ac.jp (I.E.); 3Department of Breast Surgery and Oncology, Tokyo Medical University, Tokyo 160-8402, Japan; tishik55@gmail.com; 4Department of Surgical Oncology, Graduate School of Medicine, Gifu University, 1-1 Yanagido, Gifu 501-1194, Japan; 5Department of Biostatistics & Bioinformatics, Roswell Park Comprehensive Cancer Center, Buffalo, NY 14263, USA; li.yan@roswellpark.org; 6Division of Digestive and General Surgery, Niigata University Graduate School of Medical and Dental Sciences, Niigata 951-8520, Japan; 7Department of Breast Surgery, Fukushima Medical University School of Medicine, Fukushima 960-1295, Japan; 8Department of Surgery, Jacobs School of Medicine and Biomedical Sciences, State University of New York, Buffalo, NY 14263, USA

**Keywords:** biomarker, CD4 memory T cells, CD8 T cells, cytolytic activity, immune cell, tumor immune microenvironment, tumor infiltrating lymphocyte, triple negative breast cancer, survival analysis, xCell

## Abstract

CD8 T cell is an essential component of tumor-infiltrating lymphocytes (TIL) and tumor immune microenvironment (TIME). Using the xCell CD8 T cell score of whole tumor gene expression data, we estimated these cells in total of 3837 breast cancer patients from TCGA, METABRIC and various GEO cohorts. The CD8 score correlated strongly with expression of *CD8* genes. The score was highest for triple-negative breast cancer (TNBC), and a high score was associated with high tumor immune cytolytic activity and better survival in TNBC but not other breast cancer subtypes. In TNBC, tumors with a high CD8 score had enriched expression of interferon (*IFN*)-α and *IFN*-γ response and allograft rejection gene sets, and greater infiltration of anti-cancerous immune cells. The score strongly correlated with CD4 memory T cells in TNBC, and tumors with both a high CD8 score and high CD4 memory T cell abundance had significantly better survival. Finally, a high CD8 score was significantly associated with high expression of multiple immune checkpoint molecules. In conclusion, a high CD8 T cell score is associated with better survival in TNBC, particularly when tumor CD4 memory T cells were elevated. Our findings also suggest a possible use of the score as a predictive biomarker for response to immune checkpoint therapy.

## 1. Introduction

The tumor immune microenvironment (TIME) plays a critical role in tumor progression, response to therapeutics, and prognosis in several cancers [1], including breast cancer [2,3]. Tumor-infiltrating lymphocytes (TIL) are one of the major components of TIME, and the density and types of lymphocytes in the TIL fraction of a tumor have marked prognostic associations in breast cancer [4,5]. This is particularly the case for triple-negative breast cancer (TNBC), which has the greatest number of TILs [3,5]. TNBC has a worse prognosis than other breast cancer subtypes due to its aggressive clinical behavior and lack of molecular targets for therapy [6,7]. Recently, TIL has been reported not only as a prognostic indicator but also a predictive biomarker for pathological complete response (pCR) following neoadjuvant chemotherapy (NAC), which is considered an important surrogate to predict the long-term clinical outcome for TNBC patients [5,8]. Loi et al. showed that TIL was associated with reduced risk of death in 2009 TNBC patients [5], and Denkert et al. showed a positive association between TILs and response to NAC, including the achievement of pCR in TNBC, in 1000 breast cancer patients [9]. Another study has reported similar results [10].

CD8 T cells are an important immune cell in TIL. Baker et al. reported a prognostic significance of CD8 T cells in estrogen receptor (ER)-negative breast cancer, but not in ER-positive cancer in a study of 1854 patients [8]. In addition to CD8, the importance of another subtype of immune cells such as CD4 T cell and macrophage has also been reported. For example, increased CD4 and CD8 T cell abundance in TIL is associated with better survival outcomes [11,12]. Nonetheless, the literature regarding the characterization of TIL and their prognostic utility in breast cancer has been conflicting [13,14]. One explanation for this inconsistency is the variability in the methods and criteria used to quantify TIL. TIL is typically measured by hematoxylin-eosin (H&E) staining and immunohistochemistry (IHC) for research purpose, which are the recommendations of the TILs Working Group [15,16,17]. However, inter-observer variation between examining pathologists is still deemed too large for TIL evaluation to be used in routine clinical practice [18]. Consequently, numerous studies have investigated various methodologies to quantify TIL, such as H&E assessment [9,19,20], IHC [21], and mRNA expression profiling [22,23], to identify an accurate and efficient method that can be reliably used across multiple studies.

Our group and others have reported the association of cancers with features of TIME, estimated from transcriptome data of bulk tumors by computational algorithms [24,25,26,27,28,29,30,31]. The xCell algorithm was used to estimate 64 immune and stromal cell types using gene signature profiles unique to each cell type [32]. The scores allow comparison of the estimated number of immune cells between samples, and their utility has been validated by others [33,34]. We have reported that a fibroblast score, calculated by xCell algorithm from transcriptome data [32], predicted a worse outcome in pancreatic cancer patients [35]. We also found that adipocytes in the TIME of breast cancer were associated with metastatic and inflammation-related pathways, particularly in ER-positive/human epidermal growth factor receptor 2 (HER2)-negative breast cancer [36].

Here, we hypothesize that CD8 T cells are associated with better clinical outcome, such as survival and response to treatment. We test this hypothesis using the xCell-gene-signature-based method to infer CD8 T cell abundance in the tumor of large cohorts with publicly available data—The Cancer Genome Atlas (TCGA), Molecular Taxonomy of Breast Cancer International Consortium (METABRIC), and multiple Gene Expression Omnibus (GEO) cohorts.

## 2. Results

### 2.1. CD8 T Cell Score is Associated with Clinical Aggressiveness in Breast Cancer

We analyzed the gene expression of bulk tumors using the xCell algorithm [32] to quantify CD8 T cells abundance as the xCell CD8 T cell score. The genes that xCell uses for scoring CD8 T cells are listed in Appendix A. To establish the accuracy of this scoring method, we examined if the CD8 T cell score correlates with expression of the *CD8* T cell surface marker genes, *CD8A* and *CD8B*. The CD8 score significantly correlated with the expression of both *CD8A* and *CD8B* in TCGA (Figure 1A; *CD8A*: *r* = 0.58 and *CD8B*: *r* = 0.52, both *p* < 0.01) as well as the METABRIC cohort (Figure 1A; *CD8A*: *r* = 0.86, *CD8B*: *r* = 0.51, both *p* < 0.01).

Next, we examined if the CD8 score has any relationship with the clinical aggressiveness of breast cancer. Although there was no relationship between the score and the American Joint Committee on Cancer (AJCC) stage of tumors, the CD8 score was uniformly higher among tumors of higher Nottingham pathological grade in both cohorts (Figure 1B; both *p* < 0.001). Further, when tumors were dichotomized to high and low groups using the score’s median value for the cohort, *MKI67* gene expression was higher among high compared to low CD8 score tumors in both TCGA and METABRIC cohorts (Figure 1B; both *p* < 0.001). Overall, these results suggest that the CD8 score is associated with cancer cell proliferation.

### 2.2. CD8 T Cell Score Is Highest in Triple Negative Breast Cancer (TNBC) Subtype, and TNBC Patients with High Score Have Better Survival

Since the degree of lymphocyte infiltration is known to differ between breast cancer subtypes, we investigated the distribution of the CD8 score among various subtype. In both TCGA and METABRIC cohorts, the CD8 score was significantly higher in TNBC subtype in which ER/PR/HER2 status was determined by IHC (Figure 2A; both *p* < 0.001), as well as in the basal subtype of PAM50 gene expression-based classification (*p* = 0.003, and *p* < 0.001, respectively).

We next examined the association of the CD8 score with clinical outcomes, including treatment response to NAC and Disease-Specific Survival (DSS). Surprisingly, the score was not associated with pCR rate after NAC in TNBC or in ER-positive/HER2-negative breast cancer in the GSE20194, GSE25066, and GSE32646 cohorts (Figure 2B). On the other hand, high CD8 score TNBC was significantly associated with better survival consistently in both TCGA and METABRIC cohorts (*p* = 0.025 and *p* = 0.002, respectively), whereas there was no association in either ER-positive/HER2-negative or HER2-positive subtype (Figure 2C). Taken together, these findings show that the CD8 score is highest in TNBC, and that a high CD8 score is significantly associated with better survival in TNBC, but no other subtypes. Additionally, given that TNBC is a highly proliferative subtype, it was of interest to investigate whether CD8 score can predict the survival of highly proliferative ER-positive breast cancer, including the luminal B subtype. To this end, we compared the survival by CD8 score levels in ER-positive breast cancer with high vs. low proliferation (*MKI67* expression) and luminal A vs. B in TCGA and METABRIC cohorts. As shown in Appendix A, we found that there was no significant difference in DSS in any of the groups in both cohorts (Appendix A).

### 2.3. CD8 T Cell Score Correlates with CD8A and CD8B Gene Expression, but Not Cell Proliferation in TNBC

Given that TNBC was the only subtype where CD8 score had survival relevance, it was of interest to investigate the tumor features that the CD8 score was associated with in TNBC. In both TCGA and METABRIC, the correlation of the score with *CD8A* and *CD8B* gene expression was much stronger in TNBC compared to the whole cohort (Figure 3A; *CD8A*: *r* = 0.726, and 0.896, *CD8B*: *r* = 0.694, and = 0.600, respectively, all *p* < 0.01). Unlike in the whole breast cancer cohort (Figure 1), the score was not associated with Nottingham pathological grade or *MKI67* expression in the TNBC of either cohort (Figure 3B). These results suggest that the CD8 score reflects CD8 cell infiltration strongly in TNBC; however, it does not correlate with cell proliferation in this subtype.

### 2.4. TNBC Tumors with High CD8 T Cell Score Have Elevated Immune Activity

In order to possibly obtain an insight into the biological basis for the association of a high CD8 score with improved survival in TNBC, we utilized gene set enrichment analysis (GSEA) to compare the transcriptomes of high- and low-score tumors at the level of MSigDb Hallmark gene sets. TNBC with a high CD8 score had significantly enriched expression of interferon (IFN)-α response and IFN-γ response, as well as allograft rejection gene sets in both cohorts (Figure 4A; normalized enrichment score (NES) = 1.47, NES = 1.43, and NES = 1.58; false discovery rate (FDR) = 0.16, FDR = 0.15, and FDR = 0.08 in TCGA, NES = 1.65, NES = 1.73, and NES = 1.81, all FDR < 0.001 in METABRIC cohort, respectively).

Given that TNBC tumors with a high CD8 score had enriched expression for cancer immunity-related gene sets, we investigated the relationship between the CD8 score and several immune-related features of tumors of the TCGA cohort. These features were previously quantified by Thorsson et al. [37]. The analyses revealed that TNBC tumors with a high CD8 score had significantly increased levels of leukocyte fraction, lymphocyte infiltration, TIL regional fraction, and T cell receptor (TCR), as well as B cell receptor (BCR) richness (Figure 4B, all *p* < 0.001). Furthermore, the CD8 score was highly correlated with the tumor immune cytolytic activity score (CYT), which was defined by granzyme A and perforin expression, in the TNBC of both TCGA and METABRIC cohorts (Figure 4C; *r* = 0.650 and = 0.792, respectively, both *p* < 0.01). These results suggest that the CD8 score reflected immune-cell-mediated cancer cell killing in the tumor microenvironment.

### 2.5. CD8 T Cell Score in TNBC Correlates with Infiltration by Anti-Cancer Immune Cells but not Mutation or Neoantigen Load

TIL is known to be attracted to tumors with a high mutation load because of their increased neoantigenicity [38]. To this end, we investigated whether CD8 T cell score associated with the mutation and/or neoantigen load of TNBC tumors, the measures of which were obtained from the study of Thorsson et al. [37] for the TCGA cohort. We found that the CD8 score in TNBC was not associated with any of the measurements of either mutation or neoantigen burden (Figure 5A; all *p* > 0.05).

Next, we studied the difference in the immune cell compositions of high compared to low CD8 score TNBC tumors of the TCGA and METABRIC cohorts. The xCell algorithm was used to estimate the abundance of various types of immune cells. We found that in both cohorts, high CD8 score TNBC tumors were highly infiltrated with anti-cancerous immune cells (CD4 memory T cells, M1 macrophages, and B cells) (Figure 5B; all *p* < 0.001). There was no consistent association between the CD8 score and abundance of pro-cancerous immune cells. For regulatory T cells (Tregs), T helper 2 cells (Th2), and M2 macrophages, there was no significant association in the TCGA cohort, but high CD8 score TNBC tumors of METABRIC cohort had a high amount of Treg and Th2 cells (both *p* < 0.001). These findings suggest that high CD8 score TNBC tumors have a high infiltration of anti-cancerous immune cells, even though their mutation or neoantigen load is similar to low CD8 score TNBC tumors.

### 2.6. High CD8 T Cell Score Accompanied by High CD4 Memory T Cell Infiltration Is Associated with Better Survival in TNBC

Among the anti-cancerous immune cells whose levels were increased in high CD8 score TNBC tumors, the abundance of CD4 memory T cells had a strong correlation with the score in both TCGA and METABRIC cohorts (Figure 6A; CD4 memory T cells; *r* = 0.72, and *r* = 0.80, respectively). Interestingly, high CD8 score TNBC tumors with high CD4 memory T cells had significantly better disease-free survival (DFS) and overall survival (OS) in the TCGA cohort (Figure 6B; *p* = 0.063 in DSS, *p* = 0.013 in DFS, and *p* = 0.008 in OS). This association of CD4 memory T cell abundance with survival was not observed for the low CD8 score TNBC tumors (Figure 6C; *p* = 0.301 in DSS, *p* = 0.463 in DFS, and *p* = 0.403 in OS). Thus, in TNBC, patients with a high infiltration of tumors by both CD4 memory T cells and CD8 T cells have better survival.

We also found that in both TCGA and METABRIC cohorts, TNBC tumors with a high CD8 T cell score have significantly elevated expression of multiple immune checkpoint molecules—programmed cell death 1 (*PD-1*), programmed death ligand 1 (*PD-L1*) and 2 (*PD-L2*), cytotoxic T-lymphocyte-associated protein 4 (*CTLA4)*, indoleamine dioxygenase 1 (*IDO1*) and 2 (*IDO2*), lymphocyte activation gene 3 (*LAG3*) and tyrosine-based inhibitory motif domain (*TIGIT*) (Figure 6D). These findings suggest that the CD8 score may have a potential utility as a predictive biomarker for the treatment of breast cancer with immune checkpoint inhibitors

## 3. Discussion

To examine the clinical relevance of CD8 TIL in breast cancer, we applied the xCell algorithm to bulk tumor transcriptomes to quantify the CD8 T cell score of tumors of multiple cohorts of breast cancer patients. The score correlated with the expression level of CD8 T cell-related genes, as expected, and it was highest in the TNBC subtypes of breast cancer compared to others in both the TCGA and METABRIC cohorts. A high CD8 score was significantly associated with better survival in TNBC but not in other subtypes, though the score had no association with pCR rate after NAC for any subtype. A high CD8 score TNBC was enriched for *IFN*-α and *IFN*-γ as well as allograft rejection gene sets, and had an increased immune cytolytic activity (CYT) score in both cohorts. However, the score was not associated with tumor mutation or neoantigen burden. The score strongly correlated with CD4 memory T cells in TNBC, and tumors with both a high CD8 score and high CD4 memory T cell abundance had significantly better survival. Finally, a high CD8 score was significantly associated with high expression of multiple immune checkpoint molecules.

The mainstays of analyzing immune cells are IHC and flow cytometry. However, the evaluation of immune cells by IHC may vary greatly depending on the site or observer. The flow cytometry of fresh human tumor samples in clinical setting is also difficult, and labor- and cost-intensive. With the improvement in sequencing technology in recent years, research on immune cells using whole-tumor transcriptomic data has become very common [39,40,41,42,43,44,45,46]. In the current study, the prognostic utility of CD8 score is consistent with the fact that the effect of CD8 T cells is more powerful in TNBC, and abundant TIL contribute to the better clinical outcome in TNBC [8]. Furthermore, high CD8 score TNBC tumors were also found to have high immunoreactivity.

In breast cancer, TILs are largely composed of CD8, and to a lesser extent, CD4 T cells, regulatory T cells, macrophages, mast cells, and plasma cells [47]. CD8 T cells, CD4 T cells, and regulatory T cells are the main players for immune surveillance and tolerance and they highly correlate with each other in TIME [48]. CD8 T cells lyse and clear cancer cells directly, but after being primed by CD4 T cells [11,12,49] and the crosstalk of these T lymphocytes is part of the cancer immune cycle [50]. Increased CD4 T cell infiltration has been reported to be associated with better survival outcomes, and Matsumoto et al. demonstrated that an increase in both CD4 and CD8 T cells signified good prognosis in TNBC [51]. Our current study, which used transcriptome to quantify CD8 and CD4 cells, completely echoed these results, which showed that our CD8 T cell score is a valid tool to estimate the number of CD8 cells in a clinical setting. On the other hand, regulatory T cells, which constitute 5–10% of CD4 T cells, suppress the proliferation and cytokine production of CD8 T cells [52], and Suzuki et al. reported that not the number of CD8 cells or regulatory T cells, but CD8/regulatory T cell ratio alone correlated with colorectal cancer patient survival [53]. Therefore, it may be of interest to study the clinical relevance of regulatory T cells in breast cancer in the future. Finally, high densities of tumor-associated plasma cells were reported as a predictive and prognostic factor in TNBC [54].

Although the CD8 score showed similarities with many reports, it did not associate with pCR following NAC. It is known that the intensity of the tumoral immune response influences the effectiveness of cancer therapy in TNBC [5,55]. Given the strong evidence that the amount of TIL is associated with pCR after NAC [5,9,10], we speculate that the NAC response may be related to immune cells other than CD8 T cells in TIL. Indeed, the previous studies identified TIL pathologically, and did not identify the detailed types of lymphocytes. Campbell et al. reported that the level of macrophage infiltration is predictive of NAC response [56]. To this end, it may be interesting to investigate which immune cell closely associates with response to NAC in the future. 

There was no association between CD8 score and mutation or neoantigen load in this study. It is well known that tumors with a high tumor mutation burden produce a larger number of neoantigens, making them more immunogenic [57]. On the other hand, it was argued that the excess lymphocyte infiltration in breast cancers involving BRCA1 mutations is simply an exaggeration of a phenotypic feature that has no bearing on disease progression [38]. Further investigation of the mechanisms responsible for the accumulation of lymphocytes within breast cancers with regards to tumor mutations and NAC response is warranted.

Currently, immunotherapy is only indicated for TNBC in breast cancer. After the results of Impassion 130 trial, a *PD-L1* inhibitor agent, atezolizumab, in combination with nab-paclitaxel therapy, was approved by Food and Drug Administration for patients with locally advanced or metastatic TNBC that shows *PD-L1* expression in more than 1% of any cells [58]. KEYNOTE-522 is a phase III study of NAC combined with pembrolizumab in patients with TNBC [59]. The trial reported a significantly higher pCR rate in the pembrolizumab combined group than in the chemotherapy alone group. Hypothetically, some believe that immune checkpoint inhibition on primary tumors exposes the host immune cells to abundant tumor antigens compared from adjuvant setting, thus immunotherapy may play an important role in NAC for TNBC and eventually become standard-of-care for a subset of TNBCs. However, the challenge of this field is well-defined biomarkers for better patient selection. Given that the high CD8 score TNBC tumors have significantly higher expression of multiple immune checkpoint molecules, we cannot help but speculate that our CD8 T cell score may be a useful tool as a biomarker of treatment response to immune checkpoint inhibitors.

While the current study has supplied much useful information concerning the prognostic utility of the CD8 T cells score in TNBC, the study has several limitations that we acknowledged. The study is a retrospective that used publicly available datasets. Our analyses are limited to the clinical parameters, the quality, and exact spatial location of where the sample was taken by the original authors. Furthermore, we also did not conduct any in vitro and in vivo experiments, and thus are reliant on the current literature to understand the underlying mechanisms. Finally, prospective studies to investigate whether patients selected by CD8 score respond to immune checkpoint inhibitors are needed to establish the CD8 score as a predictive biomarker.

In conclusion, a high CD8 T cell score is significantly associated with better survival in TNBC, particularly when CD4 memory T cells are elevated.

## 4. Materials and Methods

### 4.1. Tumor Immune Microenvironment Analysis

xCell algorithm [32] was used to examine whole-tumor gene expression data to score the relative abundance across tumors of 64 types of immune and stromal cells, as we previously described [30,35,36,45,46,60]. The CD8 T cell of xCell algorithm was used as the CD8 T cell score in this study in the same manner as was done for fibroblast [35] and adipocyte [36] previously. The genes that defined CD8 T cells are listed in Appendix A.

### 4.2. Clinical and Transcriptomic Data Collection for Breast Cancer Patients

In this study, 1065 patients of The Cancer Genome Atlas (TCGA)-BRCA [61], and 1903 patients of the Molecular Taxonomy of Breast Cancer International Consortium (METABRIC) [62] were included. For both cohorts, normalized and log_2_-transformed gene expression data were obtained from cBio Cancer Genomic Portal. We obtained the pathological grade data for the TCGA tumors using Text Information extraction System (TIES) Cancer Research Network [63], as described previously [64,65,66,67]. Further, in order to investigate the association of the CD8 T cell score with the treatment response for neoadjuvant chemotherapy (NAC), clinical and transcriptomic data of three cohorts—Shi et al. (GSE20194; *n* = 278) [68], Symmans et al. (GSE25066; *n* = 476) [69], and Noguchi et al. (GSE32646; *n* = 115) [70]—were obtained from the Gene Expression Omnibus (GEO) database. Probe-level expression values were summarized using mean to obtain gene expression values. Given that the patient data used in this study, TCGA, METABRIC and GEO cohorts, are all de-identified and are in the public domain, Institutional Review Board approval was waived.

### 4.3. Gene Set Enrichment Analysis

To explore signaling pathways enrichment, Gene Set Enrichment Analysis (GSEA) [71] was performed between low and high CD8 T cells score groups using GSEA Java software (https://www.gsea-msigdb.org/gsea/index.jsp version 4.0) with MSigDb Hallmark gene sets [72]. A false discovery rate (FDR) of less than 0.25 was used to deem statistical significance, as recommended by the GSEA.

### 4.4. Other Statistical Analyses

All analyses and data plotting were performed using R software (https://www.r-project.org/ version 4.0.1, R Project for Statistical Computing) and Microsoft Excel (version 16, Redmond, WA, USA) for Windows. All depicted boxplots are of Tukey type, showing medians and inter-quartile ranges. One-way analysis of variance (ANOVA) or Fisher’s exact tests were used to compare group means. The within-cohort median of the CD8 T cell score was used to divide patients into low and high groups. Survival among groups was compared using the Kaplan–Meier plot with the log-rank test. A *p* value less than 0.05 was considered statistically significant.

## 5. Conclusions

In conclusion, a high CD8 T cell score is associated with improved survival of TNBC patients, particularly in combination with CD4 memory T cells.

## Figures and Tables

**Figure 1 ijms-21-06968-f001:**
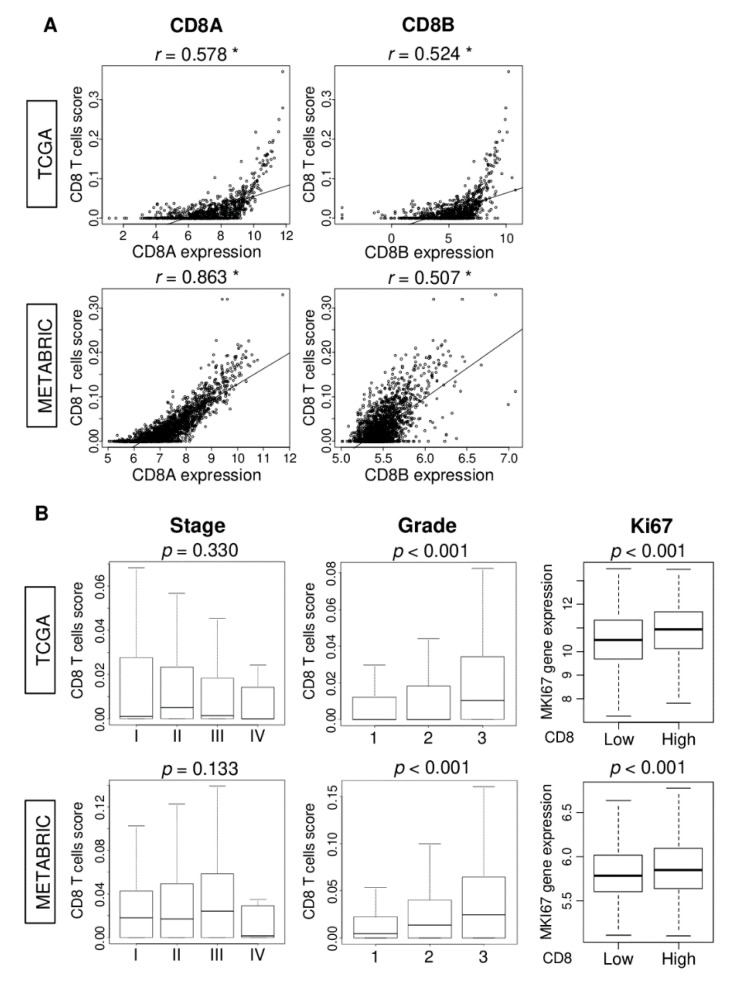
Association of CD8 T cells score with *CD8A* and *CD8B* gene expression and clinical aggressiveness in The Cancer Genome Atlas (TCGA) and Molecular Taxonomy of Breast Cancer International Consortium (METABRIC) cohorts. (**A**) Correlation between CD8 score and *CD8A* and *CD8B* gene expressions in both cohorts. Spearman correlation statistics was used for the analysis. * *p* < 0.01. (**B**) Boxplots of the CD8 score by AJCC cancer staging, Nottingham pathological grade, and comparison of low and high CD8 score groups with *MKI67* gene expression in both cohorts. One-way ANOVA was used to calculate p value. A median of the CD8 T cells score was used to divide patients into low and high score groups.

**Figure 2 ijms-21-06968-f002:**
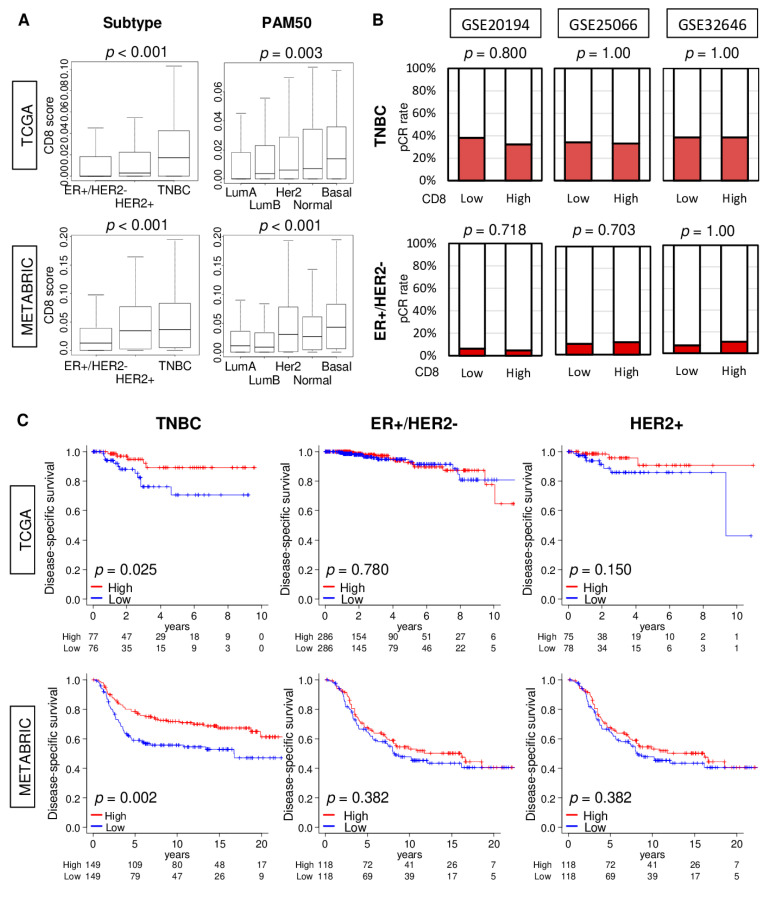
Association of CD8 T cells’ score and subtypes with neoadjuvant treatment chemotherapy (NAC) response and survival within each subtype. (**A**) Boxplots of the CD8 score by immunohistochemistry (IHC) determined subtype and PAM50 classification in the TCGA and METABRIC cohorts. One-way ANOVA was used to calculate *p* value. (**B**) Bar plots of the pathological complete response (pCR) rate between low- and high CD8 score in TNBC and estrogen receptor (ER)-positive/HER2-negative in the GSE20194 (*n* = 197), GSE25066 (*n* = 467), and GSE32646 (*n* = 115) breast cancer cohorts that underwent neoadjuvant chemotherapy (NAC). Fisher’s test was used to calculate *p* value. (**C**) Disease-Specific Survival (DSS) of CD8 score low (blue line) and high (red line) within each subtype; TNBC, ER-positive/HER2-negative, and HER2-positive, in the TCGA and METABRIC cohorts. Kaplan–Meier survival curves with log-rank test were used for the analysis.

**Figure 3 ijms-21-06968-f003:**
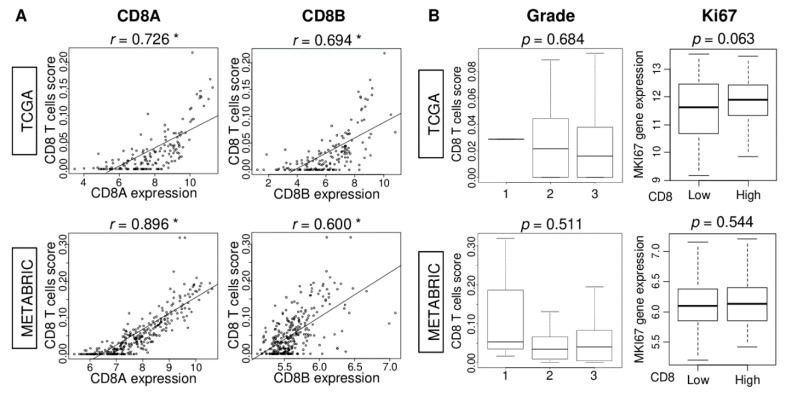
Association of CD8 T cells score with *CD8A* and *CD8B* gene expression and clinical aggressiveness of TNBC in TCGA and METABRIC cohorts. (**A**) Correlation between CD8 score and *CD8A* and *CD8B* gene expressions in TNBC of both cohorts. Spearman correlation statistics is used for the analysis. * *p* < 0.01. (**B**) Boxplots of the CD8 T cells score by Nottingham pathological grade, and comparisons between low and high CD8 score group with *MKI67* gene expression of TNBC in both cohorts. One-way ANOVA was used to calculate *p* value.

**Figure 4 ijms-21-06968-f004:**
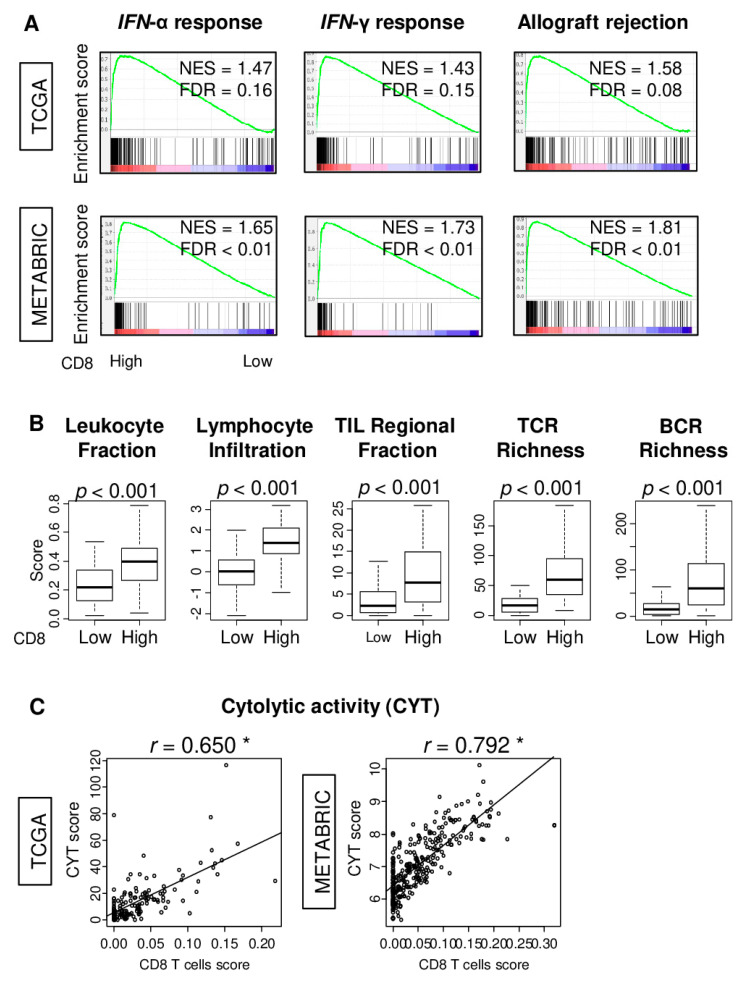
Gene Set Enrichment Analysis (GSEA) with the MSigDb Hallmark gene sets to high CD8 T cells score TNBC, association with multiple immune related scores, and correlation with cytolytic activity score (CYT) in TCGA and METABRIC cohorts. (**A**) GSEA of Hallmark IFN-α response, IFN-γ response, and Allograft rejection gene sets to high CD8 score TNBC in both cohorts, along with normalized enrichment score (NES) and false discovery rate (FDR). The statistical significance of GSEA was determined as FDR < 0.25. (**B**) Boxplots of comparison between low- and high-CD8 T cell score groups with immune-related scores; Leukocyte fraction, Lymphocyte infiltration, tumor infiltrating leukocyte (TIL) regional fraction, T cell receptor (TCR) and B cell receptor (BCR) richness score. (**C**) Correlation plots of CD8 score with CYT with spearman *r* and *p* value in both cohorts. * *p* < 0.01.

**Figure 5 ijms-21-06968-f005:**
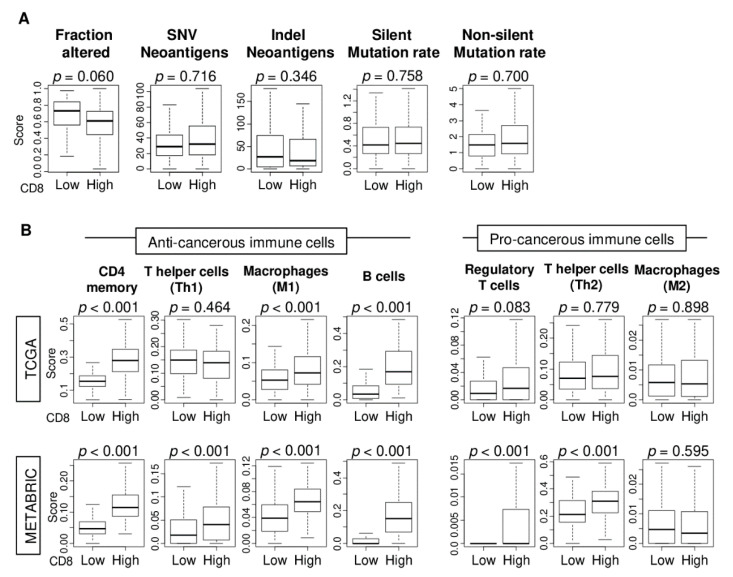
Mutation and neoantigen load as well as tumor infiltrating immune cell compositions by low and high CD8 T cells score of TNBC in the TCGA and METABRIC cohorts. Boxplots of comparison between low and high CD8 score groups with (**A**) mutation and neoantigen load scores; Fraction altered, single-nucleotide variant (SNV) and indel neoantigens, silent and non-silent mutation rate scores, and (**B**) Boxplots of comparison between low and high CD8 T cells score groups with infiltrating immune cell compositions using xCell algorithm. Anti-cancerous immune cells; CD4 memory T cells, T helper 1 cells (Th1), M1 macrophages, and B cells. Pro-cancerous immune cells; Regulatory T cells, T helper cells (Th2), and M2 macrophages. One-way ANOVA was used to calculate *p* value.

**Figure 6 ijms-21-06968-f006:**
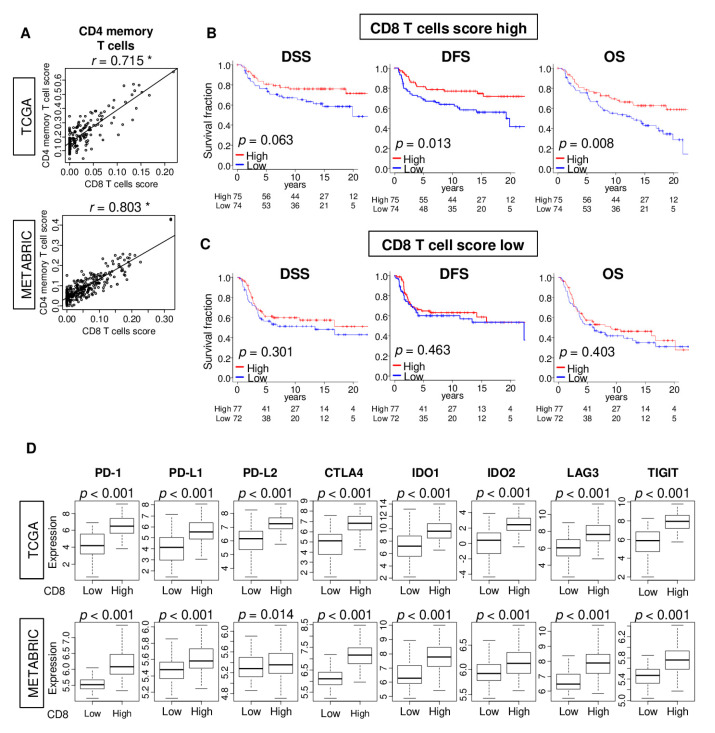
Associations of CD8 T cells score with CD4 memory T cells score and expression of immune checkpoint molecules. (**A**) Correlation plots of CD8 T cells score and CD4 memory T cells in TNBC of the TCGA and METABRIC cohorts. Spearman’s rank correlation critical value was used to the analysis. * *p* < 0.01. Kaplan–Meier plots of disease-specific (DSS), disease-free (DFS), and overall (OS) survival between low (blue) and high (red) CD4 memory T cell score groups in (**B**) high- or (**C**) low-CD8 score groups with log-rank *p* value. (**D**) Comparison of low and high CD8 T cells score groups in gene expression of immune checkpoint molecules in the TCGA and METABRIC cohorts. The one-way ANOVA was used to calculate *p* values. *CTLA4*, cytotoxic T-lymphocyte-associated protein 4; *IDO1/2*, indoleamine dioxygenase 1/2; *LAG3*, lymphocyte activation gene 3; *PD-1*, programmed death-1; *PD-L1/2,* programmed death ligand 1/2; *TIGIT*, tyrosine-based inhibitory motif domain.

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
