# Peer review of "CD8 T Cell Score as a Prognostic Biomarker for Triple Negative Breast Cancer"

_ijms, 2020, doi:10.3390/ijms21186968_

Round 1
Reviewer 1 Report
The authors hypothesized in their concise and well-written manuscript that CD8 T cells are associated with better clinical outcome. To test this hypothesis, the used the xCell gene signature-based method in large cohorts with publicly available data - The Cancer Genome Atlas (TCGA), Molecular Taxonomy of Breast Cancer International Consortium 80 (METABRIC), and three Gene Expression Omnibus (GEO) cohorts.
They showed that the expression oft he CD8 score is is higher in triple-negative breast cancer (TNBC) than in estrogen receptor (ER)-positive or HER2-positive tumors. Moreover, the CD8 score predicted improved survival only in TNBC. In a next step, the authors showed a correlation oft he CD8 score with CD8A and CD8B gene expression, but not cell proliferation in TNBC. In addition, they reported that TNBC tumors with high CD8 T cell score have elevated immune activity. Furthermore, the CD8 score correlated in TNBC with infiltration by immune cells but surprisingly neither with mutation nor neoantigen load. In addition, the authors reported that a high CD8 score accompanied by high CD4 memory T cell infiltration was associated with better survival in TNBC. Finally, they observed a significant positive association between a high CD8 score and the expression of multiple immune checkpoints like PD-1 or CTLA4.
In the discussion section of their manuscript, they place their results in the context of the existing literature and also acknowledge the potential limitations of their in silico approach, which uses only publicly available data sets.
Altogether, this retrospective study is well-conducted, interesting and thought-provoking. However, there are a few points the autors should address:
- It would be interesting if the CD8 score is also associated with improved survival in rapidly proliferating ER-positive (i.e. luminal B) breast cancer.
- It is well known that immune cells are highly correlated with each other and that not only cytotoxic CD8 cells but also CD4 cells and plasma cells representing the humoral immune system are related with an improved outcome in breast cancer. The authors should acknowledge this association briefly in their discussion.
- The conclusion that the CD8 Score could serve as a predictive biomarker for the response to immune checkpoint inhibitors is too optimistic considering that they only report an association between the CD8 Score and immune checkpoints in patients who were not treated with immune checkpoint inhibitors. The authors could put this into perspective.
Reviewer 2 Report
In this manuscript, the Authors provide an evaluation of the clinical relevance of CD8 Tumor infiltrating Lymphochytes (TILs) in breast cancer. To this aim, the Authors employs the xCell algorithm, which is a previously published and validated digital tool for dissecting the tumor microenvironment using gene expression profiles and newly generated gene signatures for 64 non-cancerous cell types. Using this approach, the Authors finds that CD8 T cell score is associated with clinical aggressiveness in breast cancer. Higher CD8 T cell score is detected in TNBC subtype; in addition, this sub-population of patients with higher CD8 T score is also characterized by better survival indexes in TNBC, but not in other molecular subtypes of breast cancer. Of note, higher CD8T score correlates with elevated immune activity, anti-cancer immune cells and CD4 memory T cells in TNBC. The Authors conclude that in the presence of high CD8 T cell score, the survival of TNBC patients is higher, particularly in combination with CD4 memory T cells.
The data are well presented and methods well validated. The manuscript is clearly written with few typos and perhaps a certain superficiality in the explanation of the xCell alogorithm in the Introduction section, which should be amended.
Author Response
In this manuscript, the Authors provide an evaluation of the clinical relevance of CD8 Tumor infiltrating Lymphochytes (TILs) in breast cancer. To this aim, the Authors employs the xCell algorithm, which is a previously published and validated digital tool for dissecting the tumor microenvironment using gene expression profiles and newly generated gene signatures for 64 non-cancerous cell types. Using this approach, the Authors finds that CD8 T cell score is associated with clinical aggressiveness in breast cancer. Higher CD8 T cell score is detected in TNBC subtype; in addition, this sub-population of patients with higher CD8 T score is also characterized by better survival indexes in TNBC, but not in other molecular subtypes of breast cancer. Of note, higher CD8T score correlates with elevated immune activity, anti-cancer immune cells and CD4 memory T cells in TNBC. The Authors conclude that in the presence of high CD8 T cell score, the survival of TNBC patients is higher, particularly in combination with CD4 memory T cells.
The data are well presented and methods well validated.
Response:
We would like to thank Reviewer #2 for taking her/his time and effort reviewing our manuscript. We are very happy to learn that Reviewer #2 found that our data were well presented and methods well validated.
The manuscript is clearly written with few typos and perhaps a certain superficiality in the explanation of the xCell alogorithm in the Introduction section, which should be amended.
Response:
We agree with the Reviewer that detailed explanation of xCell algorithm in the introduction section will improve the readers’ understanding of our paper. We have added the sentences below in the introduction section.
Introduction;
“xCell algorithm was used to estimate 64 immune and stromal cell types using gene signature profiles unique to each cell type [32]. The scores allow to compare the estimated amount of immune cells between samples, and its utility has been validated by others [33,34].”